# Genome-Wide Identification of bZIP Transcription Factors in *Cymbidium ensifolium* and Analysis of Their Expression under Low-Temperature Stress

**DOI:** 10.3390/plants13020219

**Published:** 2024-01-12

**Authors:** Huiping Lai, Mengyao Wang, Lu Yan, Caiyun Feng, Yang Tian, Xinyue Tian, Donghui Peng, Siren Lan, Yanping Zhang, Ye Ai

**Affiliations:** 1Key Laboratory of National Forestry and Grassland Administration for Orchid Conservation and Utilization, College of Landscape Architecture and Art, Fujian Agriculture and Forestry University, Fuzhou 350002, China; lhp990704@fafu.edu.cn (H.L.); wmy0818@fafu.edu.cn (M.W.); yl57@fafu.edu.cn (L.Y.); caiyun717@fafu.edu.cn (C.F.); tianyang_86@163.com (Y.T.); fjpdh@126.com (D.P.); lsr9636@163.com (S.L.); 2Anhui Province Key Laboratory of Forest Resources and Silviculture, School of Forestry and Landscape Architecture, Anhui Agricultural University, Hefei 230036, China; 18937660472@163.com

**Keywords:** bZIP, *Cymbidium ensifolium*, low-temperature stress, expression patterns

## Abstract

The basic leucine zipper (bZIP) transcription factors constitute the most widely distributed and conserved eukaryotic family. They play crucial roles in plant growth, development, and responses to both biotic and abiotic stresses, exerting strong regulatory control over the expression of downstream genes. In this study, a genome-wide characterization of the CebZIP transcription factor family was conducted using bioinformatic analysis. Various aspects, including physicochemical properties, phylogenetics, conserved structural domains, gene structures, chromosomal distribution, gene covariance relationships, promoter *cis*-acting elements, and gene expression patterns, were thoroughly analyzed. A total of 70 *CebZIP* genes were identified from the *C. ensifolium* genome, and they were randomly distributed across 18 chromosomes. The phylogenetic tree clustered them into 11 subfamilies, each exhibiting complex gene structures and conserved motifs arranged in a specific order. Nineteen pairs of duplicated genes were identified among the 70 *CebZIP* genes, with sixteen pairs affected by purifying selection. *Cis*-acting elements analysis revealed a plethora of regulatory elements associated with stress response, plant hormones, and plant growth and development. Transcriptome and qRT-PCR results demonstrated that the expression of *CebZIP* genes was universally up-regulated under low temperature conditions. However, the expression patterns varied among different members. This study provides theoretical references for identifying key *bZIP* genes in *C. ensifolium* that confer resistance to low-temperature stress, and lays the groundwork for further research into their broader biological functions.

## 1. Introduction

The basic leucine zipper (bZIP) transcription factors (TFs) are ubiquitously found in eukaryotes and are characterized by a highly conserved bZIP domain, which includes a region rich in basic amino acids and a leucine zipper [1,2,3]. Specifically, the basic region is located at the N-terminal end and exhibits a high degree of conservation, consisting of 16 amino acid residues that recognize and bind to specific sequences on the promoter through the N-X7-R/K motifs [3]. On the other hand, the leucine zipper region is relatively less conserved and is marked by a leucine or a repetitive region containing seven amino acid residues, including isoleucine, methionine, and other hydrophobic amino acids [4]. The leucine zipper can form an amphipathic α helix, mediating the formation of homodimers or heterodimers of bZIP proteins with transcriptional activation or repression functions [5,6]. In plants, bZIP proteins preferentially recognize *cis*-acting elements on promoters with ACGT core sequences, such as the G-box (CACGTG) and C-box (GACGTC) [7,8]. Notably, most of the promoter regions of *bZIP* genes induced by abscisic acid (ABA) contain these *cis*-acting elements.

bZIP TFs have been found to play a widespread role in regulating various plant life activities, including seed germination and maturation, flower development, root development, plant senescence, photomorphogenesis, and light signaling [3,9,10,11,12,13,14]. The co-expression of *AtbZIP10*/*25* with ABI3 significantly enhances the activation of the *At2S1* (2S storage protein) promoter, indicating their involvement in the composition of the regulatory complex associated with seed maturation-specific expression [15]. In addition, AtbZIP18 interacts with AtbZIP34, while AtbZIP52 interacts with AtbZIP61 in yeast to co-regulate pollen development [10]. Notably, the mutants *AtbZIP18* and *AtbZIP34* exhibit a failure in forming normal pollen walls, leading to a reduced amount of viable pollen [10]. In *Zea mays*, *ZmbZIP4* binds to the promoters of root development-related genes, such as *ZmLRP1* and *ZmSCR*, thereby activating the transcription of root development-related genes, consequently promoting root growth and development [12].

Furthermore, bZIP TFs are involved in abiotic stresses, including low-temperature stress, drought stress, and salinity stress, as well as in biotic stresses such as disease and pathogen defense, and they also contribute to the induction of various hormones [4,16,17,18,19,20,21]. For instance, the overexpression of *TabZIP6* in *Arabidopsis thaliana* significantly reduced the expression of genes like *CBF* and *COR*s in plants, leading to decreased freezing resistance in transgenic *A. thaliana* seedlings [22]. In *Camellia sinense*, *CsbZIP6* acts as a negative regulator of the low-temperature stress responses, reducing the expression of low-temperature-responsive genes and influencing ABA sensitivity [23]. Moreover, the overexpression of *BnbZIP2* in *Arabidopsis* significantly increases the sensitivity of transgenic plants to drought and salinity stress [24]. Under drought stress, the down-regulation of ABA biosynthesis and signal transduction-related genes in tomato occurred when *SlbZIP1* was silenced, indicating that *SlbZIP1* plays a role in promoting drought resistance in tomato [25].

*Cymbidium ensifolium* thrives in warm and humid environments, displaying limited cold tolerance. Safe overwintering requires temperatures above 5 °C, and the plant becomes susceptible to frost damage when temperatures drop below 0 °C. Despite bZIP TFs having been identified in various species, including *A. thaliana* (78 genes) [7], *Glycine max* (160 genes) [18], *Ipomoea batatas* (87 genes) [26], *Sorghum bicolor* (92 genes) [27], *Juglans regia* (88 genes) [28], *Populus trichocarpa* (41 genes) [29], and *Malus pumila* (112 genes) [30], their presence is mainly concentrated in cash crops. The bZIP gene family has been rarely reported in *C. ensifolium*. This study is based on the genome-wide sequencing data of *C. ensifolium*, aiming to identify and characterize the members of the bZIP family in *C. ensifolium* (CebZIP). Bioinformatic methods were applied to comprehensively analyze the protein properties, phylogenetic relationships, conserved motifs, gene structure, and other information of the CebZIP family. The findings of this study can provide reference for enhancing the cold tolerance of *C. ensifolium*, and provide a theoretical foundation for the functional verification of *CebZIP* genes (*CebZIPs*) and the identification of key response genes in future research.

## 2. Results

### 2.1. Identification and Characterization Analysis of CebZIPs in C. ensifolium

Using HMM from the Pfam database for alignment on the *C. ensifolium* genome database, 78 matches were obtained as candidate CebZIP proteins [31]. By identifying the complete bZIP domains in both the Pfam and SMART databases, a total of 70 CebZIP protein sequences were finally selected [31,32,33]. These CebZIP proteins were designated CebZIP1–CebZIP70 in accordance with their chromosomal arrangement (Appendix A). An analysis of the physicochemical properties revealed that the molecular weights (MW) of the 70 CebZIP proteins ranged from 9.04 kDa (CebZIP31) to 75.16 kDa (CebZIP67). The smallest CebZIP protein was CebZIP23 (77 aa and 234 bp), while the largest was CebZIP67 (697 aa and 2094 bp). The isoelectric point (PI) ranged from 4.99 (CebZIP10) to 11.55 (CebZIP45). The instability coefficients ranged from 29.74 (CebZIP3) to 86.21 (CebZIP45), and the aliphatic index coefficients ranged from 53.46 (CebZIP8) to 90.17 (CebZIP55). The grand average of hydropathicity (GRAVY) consistently remained below 0 for all CebZIP proteins, indicating their hydrophilic nature.

### 2.2. Phylogenetic Analysis of CebZIPs

*A. thaliana* is one of the earliest plants studied for the bZIP TF family, holds significant reference value, and is commonly employed as a model genetic plant for plant gene research [3,7]. To gain a deeper understanding of the evolutionary properties of CebZIP family members, a phylogenetic tree was constructed by combining 70 *CebZIP*s with 78 *AtbZIP*s (Figure 1) [7]. Following the classification method used for AtbZIP proteins in previous studies [7], the *CebZIP*s were clustered into 11 subfamilies (A–I, K, and S), along with four unclassified genes. In the AtbZIP family, the largest subfamily is subfamily S (17 members), and the smallest subfamilies are subfamilies K and J (each with 1 member). Similarly, in the CebZIP family, the number of members varies among the 11 subfamilies, with the largest number in subfamily S (18 members) and the smallest in subfamily K (1 member). The high amino acid sequence similarity observed among members of the same subfamily demonstrates substantial homology between *CebZIP*s and *AtbZIP*s.

### 2.3. Analysis of Conserved Domains, Conserved Motifs, and the Gene Structure of the CebZIPs

To further study and characterize the homologous conserved domains in CebZIP proteins, a multiple sequence alignment and a weblogo analysis of CebZIP proteins were performed by analyzing the characteristic amino acid sequences of the leucine zipper and basic domains within the bZIP gene family. Similar to other species, CebZIP proteins contain basic domains and leucine zipper, namely, [-N-X7-R/K-] and [-L-X6-L-X6-L-] (Figure 2).

Predictions of conserved motifs among the 70 CebZIP proteins revealed the presence of 10 motifs, strategically distributed at the N-terminal and C-terminal ends, and the middle part of the proteins. Motif 1 displayed universal presence across all CebZIP proteins, underscoring its high conservation within the CebZIP gene family. Moreover, certain motifs have significant specificity. For instance, motifs 2, 5, 7, and 8 were exclusive to subfamily D, while motif 4 was uniquely present in subfamily I. As expected, substantial differences in motif composition were observed between subfamilies. In contrast, motifs within the same subfamily demonstrated similarities, and the overall pattern of motif distribution within each subfamily remained essentially similar.

To further verify the structural characteristics of all *CebZIP*s, we analyzed the number and structure of intron/exon (Figure 3). The gene structure of *CebZIPs* displayed significant variability and diversity in terms of the relative position and number of introns and exons. The intron numbers among the 70 *CebZIP*s ranged from 0 and 18. Seventeen (24.3%) *CebZIP*s showed no intron structure, with the majority concentrated in the S subfamily. *CebZIP53*, belonging to the G subfamily, exhibited the highest number of introns (18), resembling the gene structure of *AtbZIP*s. The exon numbers varied from two to seven in subfamilies A, B, C, E, F, H, I, and K. Conversely, subfamilies D and G have the most complex gene structures, with exon numbers ranging from nine to nineteen.

### 2.4. Chromosomal Distribution and Covariance Analysis of CebZIP Family Members

A chromosomal distribution map of CebZIP gene family was generated using Tbtools [34]. The *C. ensifolium* genome comprises 20 chromosomes, and the 70 *CebZIP*s are randomly distributed across 18 of these chromosomes, with chromosomes 11 and 18 lacking *bZIP* genes (Figure 4). Chromosome 13 accommodates the highest number of *CebZIP* genes (eight), while chromosomes 9, 12, 15, 16, 17, and 19 each contain two genes.

Gene duplication is a major driver in the evolution of genomes and genetic systems. The analysis of the intraspecific collinearity in *C. ensifolium* revealed that the CebZIP family consists of 19 duplicated gene pairs, including 9 pairs of segmental duplicates and 10 pairs of tandem duplicates (Figure 5). Chromosomes 1 harbors the highest number of tandem duplicated gene pairs (three), followed by chromosomes 13, 4, 6, 14, and 19 (two each), while chromosomes 20 contained only one pair. To explore the evolutionary characteristics of the bZIP family in orchids, collinearity relationships were established by comparing *C. ensifolium* with two other orchids, *Cymbidium goeringii* and *Dendrobium chrysotoxum* (Figure 6). The number of homologous pairs between *C. ensifolium* and these two orchids was similar, with 56 and 54 pairs, respectively.

To investigate the environmental selection pressure experienced by these duplicated *CebZIP* genes, we calculated the non-synonymous site (*K*a) and synonymous site (*K*s) of each gene pair and analyzed their ratios (*Ka*/*Ks*) (Appendix A). Three pairs of *CebZIP* genes yielded NaN (Not a Number), which might be attributed to potential genome sequencing errors. Among the remaining 16 duplicated gene pairs, 2 gene pairs displayed only synonymous without non-synonymous. The remaining 14 *Ka*/*Ks* values were all less than 1, with 75% of them being less than 0.5, suggesting that the majority of the *CebZIP*s evolved under strong purifying selection.

### 2.5. Prediction of Cis-Acting Elements in CebZIP Promoters

To predict the biological function of the *CebZIP*s, a *cis*-acting element analysis of the 2000 bp promoter region upstream of the *CebZIP*s start codons was performed using the PlantCARE database (Figure 7) [35]. Abundant *cis*-acting elements were identified, and 19 crucial elements, excluding general transcriptional regulatory and functionally unknown elements, were selected for analysis. These included light-responsive element (e.g., G-Box, GT1-motif, and Sp1), ABA-responsive element (ABRE), and low-temperature response element (LTR), among others. These elements were categorized into three groups: plant hormones, abiotic stress, and plant growth/development. Further analysis revealed that the *cis*-acting element number in the abiotic stress group was the largest, followed by the plant hormone group. Notably, 28 (40%) genes in the CebZIP gene family contained LTR. The presence of various *cis*-acting elements suggests a wide range of functional roles for *CebZIP* genes.

### 2.6. Analysis of Expression Patterns of CebZIP Genes from Transcriptome Data

Gene expression patterns can unveil various biological functions in plants. To validate the role of *CebZIP*s in response to low-temperature stress, we used transcriptome data to examine the expression levels of *CebZIP*s at 0 h, 4 h, 12 h, and 24 h post treatment with low-temperature stress. Subsequently, we constructed a gene expression heat map (Figure 8). Following low-temperature stress, the expression level of most *CebZIP* genes changed significantly and exhibited diverse temporal expression patterns, suggesting their distinct regulatory roles in response to low-temperature stress.

After low-temperature treatment, *CebZIP2*, *CebZIP23*, *CebZIP48*, *CebZIP9*, *CebZIP66*, and *CebZIP18* displayed lower expression than the control group (0 h). Seventeen genes were up-regulated after 4 h of low-temperature stress, while twenty-seven genes showed elevated expression after 12 h of low-temperature stress. Fourteen genes exhibited high expression after 24 h of low-temperature stress. These findings underscore the functional diversity of the CebZIP gene family in responding to adverse stress in *C. ensifolium*.

### 2.7. Analysis of CebZIP Gene Expression Patterns under Cold Stress

Combining the results of *CebZIP* gene promoter and transcriptome data analysis, nine genes were selected for qRT-PCR experiment. As shown in Figure 9, their expression levels were generally consistent with the transcriptome analysis results. Under low-temperature stress, the gene expression patterns changed significantly over time. *CebZIP8* expression began to rise at 12 h, reaching its peak at 24 h. *CebZIP24* showed its highest expression at 12 h, followed by a sharp decrease. *CebZIP26* exhibited high expression at 4 h, followed by a decrease and then a slight increase at 24 h. The overall expression patterns of *CebZIP28*, *CebZIP38*, and *CebZIP70* followed an increase and a decrease, reaching a peak at 12 h, followed by a sharp down-regulation. The highest expression of *CebZIP37* and *CebZIP56* occurred at 4 h. *CebZIP43* demonstrated a slight increase at 4 h, reaching its peak at 24 h.

## 3. Discussion

*C. ensifolium* is a shade-tolerant plant typically found in cool environments such as thickets and forests, displaying low tolerance to both cold and heat. The bZIP gene family plays an essential role in regulating growth and development throughout the entire life cycle of plant, and it also contributes significantly to enhance resistance to low temperatures [36,37]. Previous research has confirmed that bZIP TFs have a strong response to various abiotic stresses, and their high expression has been shown to enhance resistance in different species, including maize [38], cotton [39], rice [40], and other crops. The genome-wide characterization of the bZIP gene family has been conducted in many species. However, it has not yet been reported for *C. ensifolium* nor has the involvement of *bZIP* genes in responding to low temperatures in *C. ensifolium* been studied.

Based on the whole genome data, we used bioinformatic methods to identify and analyze the members of the bZIP gene family in *C. ensifolium* [41]. A total of 70 *CebZIP*s were identified after removing redundancy and were found to be randomly distributed on 18 chromosomes. Using these *CebZIP* genes, we constructed an evolutionary tree together with *AtbZIP* genes. The *CebZIP*s were clustered into 11 subfamilies (A–I, K, S), a pattern observed in other species, such as sweet potato [26], tobacco [42], and pineapple [43], suggesting that the *bZIP* gene family is conserved. No *CebZIP* family members were categorized into subfamily M and subfamily J, which is similar to poplar [29]. In addition, four CebZIP proteins were not clustered and may represent new members formed during the evolutionary process, which requires further investigation of their biological functions. These findings indicated the absence or increase of bZIP members during the evolution of *C. ensifolium*, potentially linked to the diverse functions of CebZIP proteins.

The multiple sequence alignment and weblogo analysis of CebZIP proteins revealed that all CebZIP proteins contain the basic domain [-N-X7-R/K-] and leucine zipper region [-L-(X6)-L-(X6)-L-], similar to *A. thaliana*, poplar, and other species [3,29]. These results indicate that the bZIP family is highly conserved during the evolutionary process. The motif composition of the *CebZIP* varied among subfamilies, but motifs were similar within the same subfamily. Moreover, the motifs encoding the bZIP domains were relatively conserved, suggesting inter-clade-specific functions of CebZIP proteins. There also appear to be potential functional similarities among proteins within the same subfamily of *CebZIP*s.

Conservative gene structure changes are likely to represent key events in evolution, and variations in exon-intron positions and numbers can lead to specific gene functions and evolutionary directions [44]. The gene structure of *CebZIP*s is significantly diverse in terms of the relative position and number of introns/exons. The intron numbers ranged from 0 to 18, and there were notable differences in gene length. Seventeen (24.3%) *CebZIP* genes had no intron structure, with most of them concentrated in subfamily S. The high degree of similarity in the structural features of the genes may be attributed to gene duplication events during the gene family expansion [37]. Differences in gene structure arise from mutations in introns such as base substitutions, insertions, and deletions, which alter the gene sequence and result in diversity in the number and arrangement of introns, contributing to the functional development of CebZIP proteins [45].

Tandem duplication and segmental duplication are primary mechanisms responsible for the expansion of gene families and the emergence of new functional genes during gene evolution [46]. In the CebZIP gene family, a total of 19 pairs of duplicated genes were identified, including 9 pairs of segmental duplicates and 10 pairs of tandem duplicates. This suggest that gene duplication events play an essential role in the evolutionary dynamics of bZIP. In comparison, poplar has 31 pairs of segmental duplicates [29], wheat has 14 pairs of segmental duplicates [47], and tobacco has 16 pairs of segmental duplicates [42], none of which involve tandem events. In tomato, 21 pairs of segmental duplicate genes and 8 pairs of tandem duplicate genes were identified [48]. These findings underscore the ongoing evolution of the bZIP gene family, generating genes with novel functions.

The analysis of the collinearity between *C. ensifolium* and two other orchids (*D. chrysotoxum* and *C. goeringii*) revealed strong homology and highly conserved evolutionary relationships among these three orchids. To assess selection pressure on protein-coding genes [49], a selection pressure analysis was performed. The segregation and selection processes were examined by calculating *K*a/*K*s values for 19 pairs of duplicate genes. The *K*a/*K*s values of 16 pairs of duplicate genes were all < 1, with 75% of them < 0.5. These findings suggest that these genes underwent strong selection for purification, and the *bZIP* genes exhibited a slow and highly conserved evolutionary process in *C. ensifolium*, demonstrating a certain level of stability in both structure and function.

The analysis of *cis*-acting elements situated in the *CebZIP* promoter region revealed the presence of various phytohormone response elements, in addition to abiotic stress response elements. This suggests that *CebZIP*s are involved in both phytohormone and abiotic stress response pathways, which is consistent with the results observed in wheat [47]. The data indicated that *bZIP* genes across different species contain response elements for low temperature, hormones, hypoxia, and trauma, showcasing their adaptability to diverse environment conditions.

The qRT-PCR results demonstrated substantial changes in gene expression upon low-temperature induction, aligning with the findings from the transcriptome data analysis. However, different expression patterns were observed among different members, with varying maximum values and time points to reach those maxima. Specifically, *CebZIP26*, *CebZIP37*, and *CebZIP56* were highly expressed after 4 h of low-temperature stress, while *CebZIP24*, *CebZIP28*, *CebZIP38*, and *CebZIP70* peaked after 12 h. On the other hand, *CebZIP8* and *CebZIP43* were most strongly expressed at 24 h. The expression of *CebZIP26*, *CebZIP37*, and *CebZIP56* remained high after 4 h of low-temperature stress. These findings suggest that *CebZIP*s may play a significant role in the process of low-temperature resistance, demonstrating diverse low-temperature stress response modes and distinct functions in the physiological processes of *C. ensifolium* under low-temperature stress.

## 4. Materials and Methods

### 4.1. Plant Materials

The study utilized plants from the Forest Orchid Garden at Fujian Agriculture and Forestry University, all of which were grown in the same conditions within the greenhouses. Prior to the experiment, these plants underwent pre-cultivation in an artificial climate chamber, followed by a seven-day period of cultivation under standard conditions. After pre-cultivation, the plant materials were exposed to a 4 °C artificial climate incubator for 0 h, 4 h, 12 h, and 24 h. Subsequently, the leaves were aseptically cut using enzyme-free scissors, with the midrib removed. Rapid freezing with liquid nitrogen and storage in a −80 °C refrigerator ensured the preservation of the samples. All samples were subjected to three biological replicates.

### 4.2. CebZIP Transcription Factor Identification and Sequence Retrieval

The whole genome data of *C. ensifolium* were acquired from the National Genomics Data Center (https://ngdc.cncb.ac.cn/, accessed on 2 March 2023), and *Arabidopsis* bZIP TF sequences were retrieved from the TAIR database (http://www.arabidopsis.org/, accessed on 2 March 2023). The conserved domain of the bZIP transcription factor (ID: PF00170, bZIP_1) was downloaded from the Pfam database (http://pfam.xfam.org/, accessed on 2 March 2023) [31]. This conserved domain was employed as a seed model for initial screening (E-value ≤ 10^−5^) using the Simple HMM Search of the TBtools v1.120 [34]. The candidate bZIP protein sequences of *C. ensifolium* underwent structural domain detection via the SMART (http://smart.embl.de, accessed on 2 March 2023), pfam, and NCBI CDD (https://www.ncbi.nlm.nih.gov/Structure/cdd/wrpsb.cgi, accessed on 2 March 2023) databases. The removal of duplicate sequences yielded a final set of 70 putative genes [31,32,33]. The online website ExPASy was utilized to predict the molecular weight (MW), amino acid number (aa), isoelectric point (PI), aliphatic index, grand average of hydropathicity (GRAVY), and instability index of CebZIP proteins [50].

### 4.3. Phylogenetic Analysis of CebZIP Transcription Factors

The bZIP protein sequences from *C. ensifolium* and *A. thaliana* were subjected to ClustalW multiple alignment using MEGA11.0 software. The phylogenetic tree was constructed using the Neighbor-Joining (NJ) method [51]. The calibration parameter Bootstrap repetitions was set to 1000. The evolutionary tree was refined and polished using ITOL (https://itol.embl.de/itol.cgi, accessed on 11 March 2023) [52].

### 4.4. Analysis of Gene Conserved Structural Domains, Gene Structure, and Conserved Motifs

The 70 identified *CebZIP* family members were individually aligned using Clustal W, and the online software WebLogo 3 (http://weblogo.berkeley.edu/logo.cgi/, accessed on 31 March 2023) was utilized to map the conserved structural domains LOGO [53]. The gene structures (introns-exons) were analyzed and visualized using Tbtools v1.120 [34]. The online software MEME (http://meme-suite.org/tools/meme, accessed on 31 March 2023) was used to predict the conserved motifs of the *CebZIP* family [54]. The output value was set to 10, and the analysis results were visualized using Tbtools v1.120 [34]. The *K*a/*K*s ratio was calculated using Tbtools v1.120 to determine the evolutionary selection pressure of genes. Generally, if *K*a > *K*s or *K*a/*K*s > 1, the gene undergoes positive selection; if *K*a = *K*s or *K*a/*K*s = 1, the gene is subject to neutral evolution; and if *K*a < *K*s or *K*a/*K*s < 1, the gene undergoes purifying selection [34,49].

### 4.5. Chromosome Localization and Collinear Analysis of CebZIP

The chromosomal details of the 70 *CebZIP* genes were obtained from genome annotation files, and their chromosomal distribution was visualized using Tbtools v1.120 [34]. One Step MCScanX in TBtools v1.120 was used to analyze the intra-species collinear relationship of *C. ensifolium*, generating a Circos plot to represent the gene interactions [34]. The duplication events of *bZIP* genes among three species (*C. ensifolium* and *D. chrysotoxum*, and *C. goeringii*) were analyzed to obtain the collinear relationship among genes and plot the interspecies collinear map.

### 4.6. Examination of Cis-Acting Elements within the Promoter Region of CebZIP

Tbtools v1.120 was used to obtain 2000 base pair (bp) upstream of the coding sequence (CDS) of *CebZIP* genes from the genome sequence of *C. ensifolium*. The obtained sequence was submitted to the online website PlantCARE (http://bioinformatics.psb.ugent.be/webtools/plantcare/html/, accessed on 20 March 2023) for *cis*-acting element prediction [34,35]. General transcriptional regulatory elements and elements of unknown function were filtered out. The predictions were organized, classified, and visualized using EXCEL 2019.

### 4.7. Gene Expression Analysis under Low-Temperature Stress

The leaves, cultured for 0 h, 4 h, 12 h and 24 h in a 4 °C artificial climate chamber, were used for RNA extraction using a total RNA extraction kit (OMEGA, Norcross, GA, USA). Transcriptome data library construction was performed using Novozymes (Beijing, China) on the Illumina Hiseq 2500 platform. Gene expression levels are indicated by FPKM (Fragment Per Kilobase of exon model per Million mapped fragments) (Appendix A). The heatmap of *CebZIP* genes expression under different low-temperature stress treatments was generated using the HeatMap tool in TBtools v1.120 [34]. The clustering method was set to cluster rows, and the normalization mode was selected as row scale. The scale method was set to zero to one, with other parameters using the default values.

cDNA was synthesized using a reverse transcription kit. Primers were designed using Primer 5 with *GAPDH* gene as the house-keeping gene (Appendix A). Fluorescence quantification experiments were performed on an ABI 7500 real-time system (Applied Biosystems, Foster City, CA, USA) using the Hieff^®^ qPCR SYBR Green Master Mix (No Rox) (Next Sense Bio, Shanghai, China) kit. Three biological replicates and three technical replicates were performed. The 2^−ΔΔCt^ formula was used for the calculation of relative gene expression [55]. The details of the 2^−ΔΔCt^ formula and qRT-PCR data are provided in Appendix A.

## 5. Conclusions

In this study, 70 *CebZIP* genes were identified in the genome of *C. ensifolium* and their physicochemical properties were analyzed. *CebZIP* members exhibit diverse structures and have been classified into 11 subfamilies based on phylogenetic trees drawn from amino acid sequences. The conserved motifs vary among the subfamilies of the CebZIP family, while the motifs of bZIP domains encoded by the same subfamily are relatively conserved and similar. A total of 19 pairs of duplicated genes were identified in the CebZIP gene family, indicating that gene duplication events play a crucial role in driving bZIP evolution. Moreover, the expression profiling and qRT-PCR experiment showed that *CebZIP*s were subjected to low-temperature induction with varying degrees of changes in expression. These findings provide a certain theoretical foundation for identifying potential genetic resources for the molecular breeding of orchid stress resistance.

## Figures and Tables

**Figure 1 plants-13-00219-f001:**
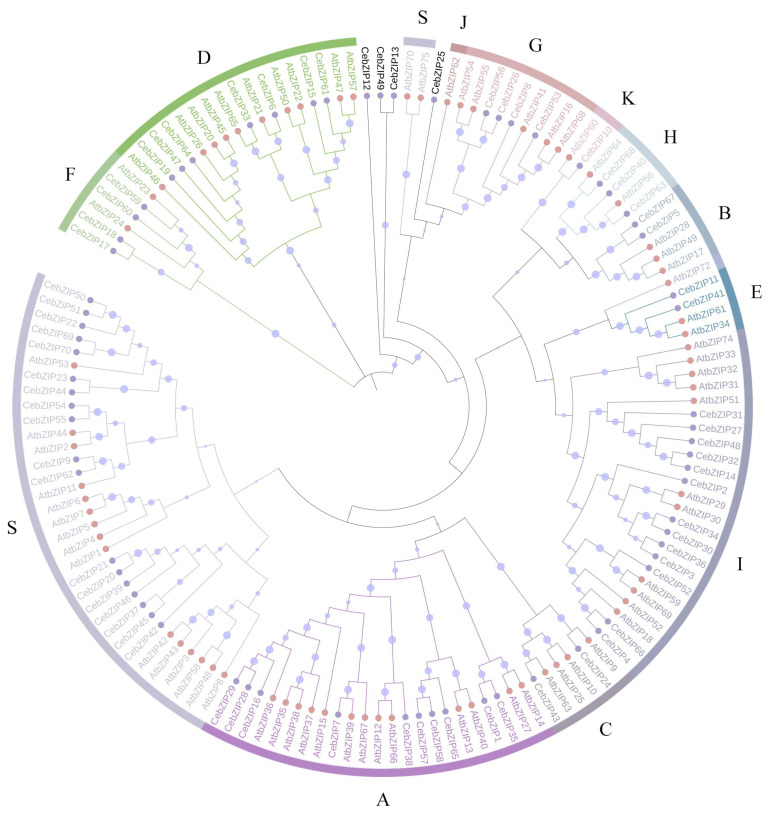
Phylogenetic tree of bZIP proteins from *C. ensifolium* and *A. thaliana*. Different bZIP subfamilies are marked by different colors. Pink and purple circles represent CebZIP proteins and AtbZIP proteins, respectively. The size of the purple circle on the branch reflects the level of support.

**Figure 2 plants-13-00219-f002:**
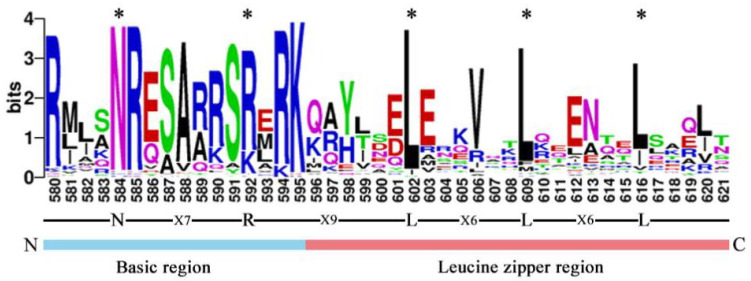
The bZIP domain of CebZIP proteins, consisting of a basic region and a leucine zipper region. The * represent the conserved amino acids of bZIP domain.

**Figure 3 plants-13-00219-f003:**
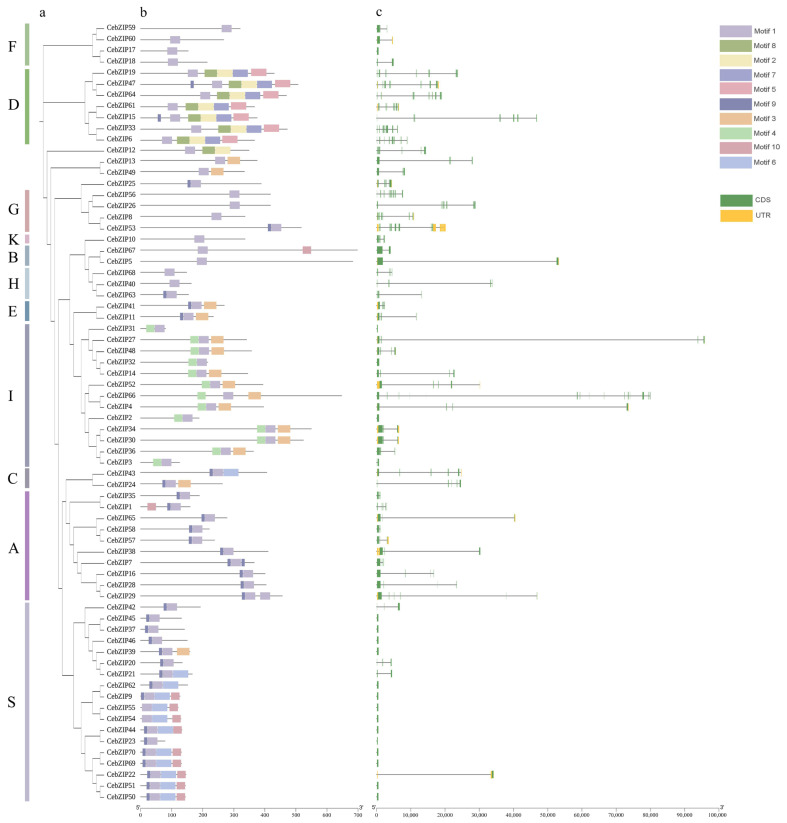
Conserved motif and gene structure analysis of *CebZIP*s. (**a**) Phylogenetic tree. (**b**) Conserved motifs are shown by colored boxes. (**c**) Exon/intron structure of *CebZIP*s are displayed by green bars and black lines, respectively.

**Figure 4 plants-13-00219-f004:**
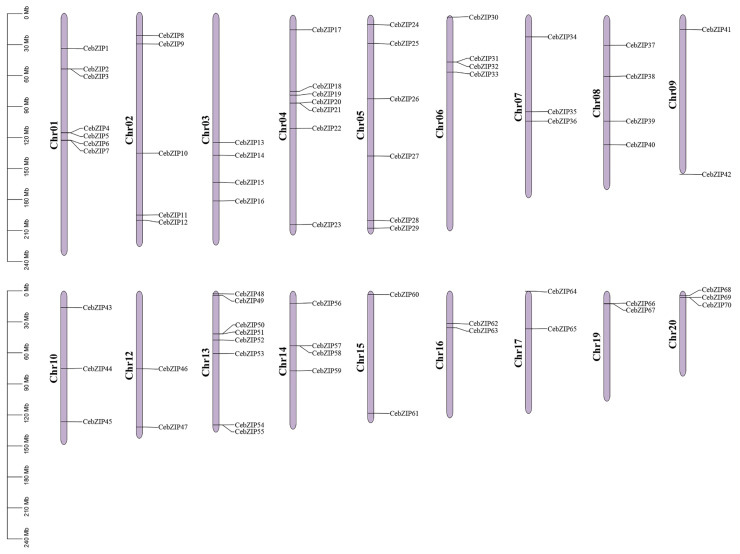
Localization and distribution of the 70 *CebZIP* genes on the 18 chromosomes of *C. ensifolium*. Red boxes indicate tandem duplicated genes. The scale represents the chromosomal distances (Mbp).

**Figure 5 plants-13-00219-f005:**
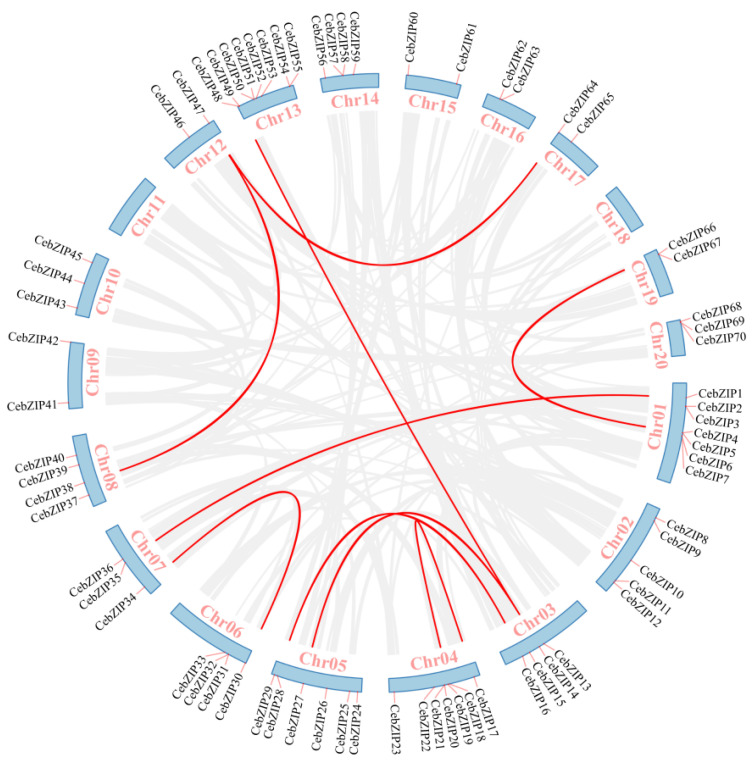
Collinear distribution of *CebZIP* genes on chromosomes. The 18 chromosomes of *C. ensifolium* are arranged in a circular blue bar. Red lines connecting chromosomes represent segmentally duplicated genes.

**Figure 6 plants-13-00219-f006:**
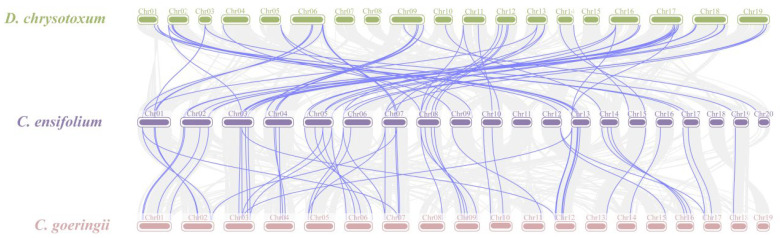
Collinearity analysis of *bZIP* genes among three orchids. Gray lines indicate all syntenic blocks between *C. ensifolium* and the other two orchids. The blue lines delineate collinear pairs of *bZIP*s.

**Figure 7 plants-13-00219-f007:**
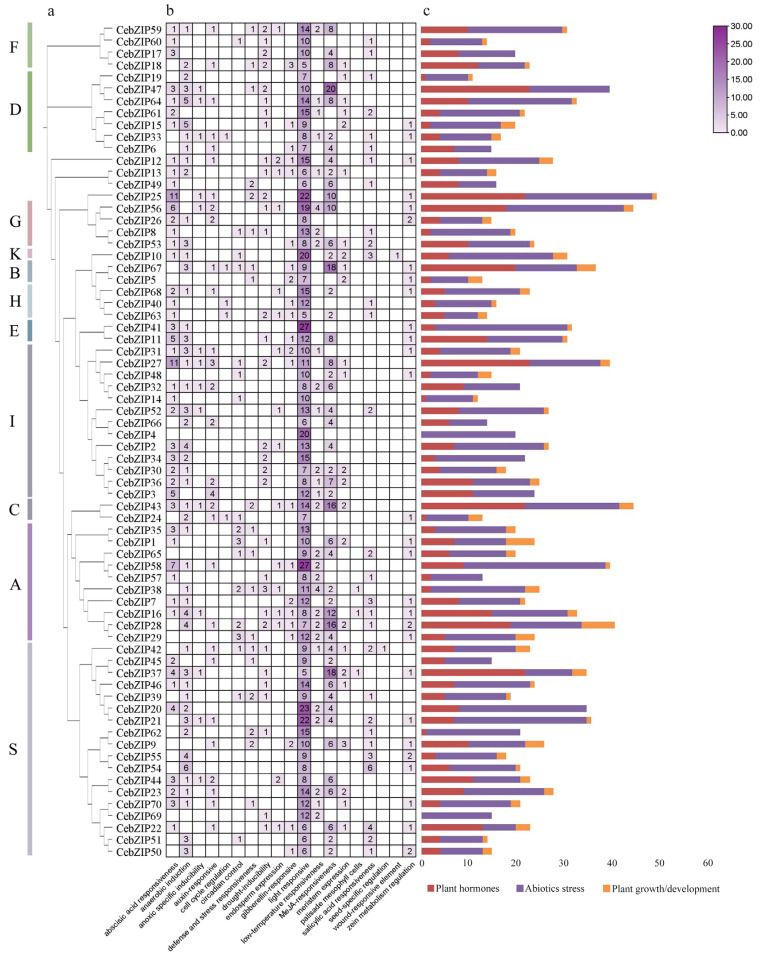
The *cis*-acting elements of *CebZIP* gene promoters. (**a**) Phylogenetic tree. (**b**) The number of *cis*-acting elements is shown in boxes. (**c**) The different colors delineate the main categories of the functional components. The size of the box delineates the number of functional components.

**Figure 8 plants-13-00219-f008:**
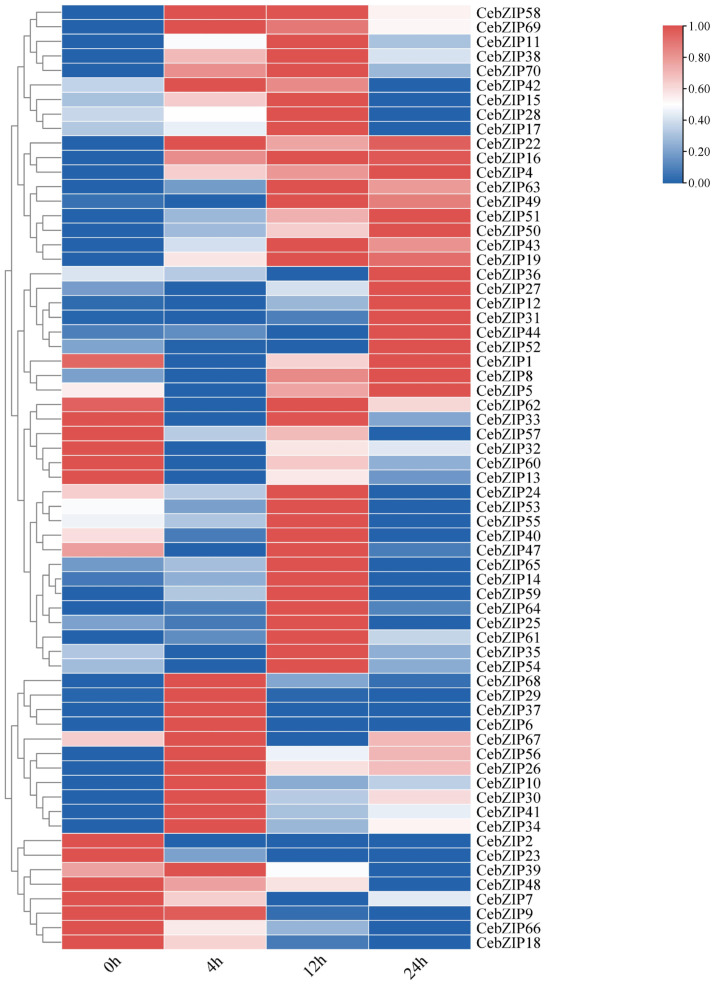
Heat map of *bZIP*s expression under low-temperature stress in *C. ensifolium*, with 0 h, 4 h, 12 h, and 24 h delineating different times of low-temperature treatment.

**Figure 9 plants-13-00219-f009:**
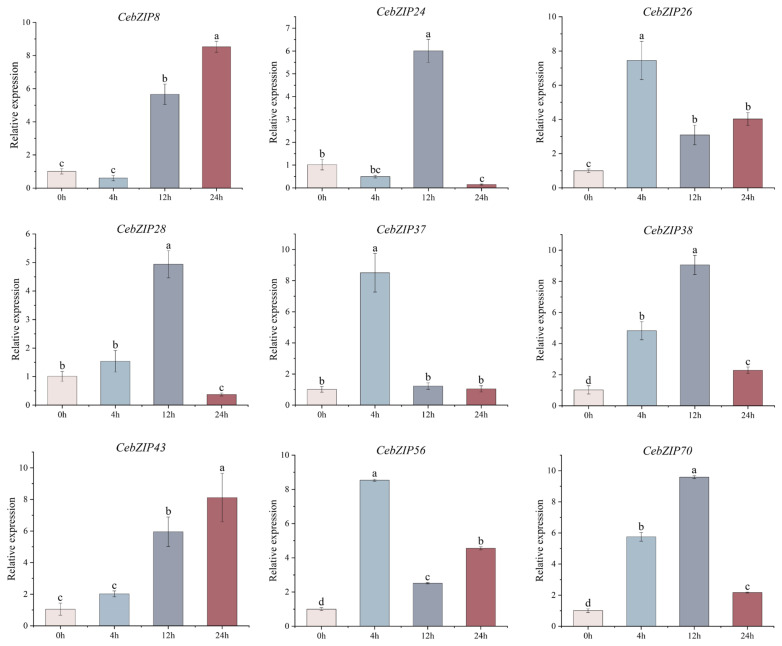
qRT-PCR analysis of nine *CebZIP* genes in *C. ensifolium* under low-temperature treatment. Error bars indicate the SD of three biological replicates. Letters delineate significant differences among treatments (*p* < 0.05, Duncan).

## Data Availability

All supporting data of the current study are available within the paper and within the Appendix A published online. The sequences of *C. ensifolium* used in this study are available at the National Genomics Data Center (NGDC). The RNA-Seq data have been deposited in NCBI under SRA accession codes: PRJNA1044247.

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
