# Peer review of "Genome-Wide Identification of bZIP Transcription Factors in Cymbidium ensifolium and Analysis of Their Expression under Low-Temperature Stress"

_plants, 2024, doi:10.3390/plants13020219_

Round 1
Reviewer 1 Report
Comments and Suggestions for Authors
In this study, authors comprehensively identified bZIP TFs in Cymbidium ensifolium and characterize them. It is important study to provide useful information that can be utilized for the future elucidation of functional diversity of bZIP TFs in this species. However, I still would like to suggest modifications of this manuscripts on the points below. Basically, these modifications could help readers to predict function of CebZIPs.
For the explanation of phylogenic analysis (P3), authors should describe which Arabidopsis bZIPs are included in the largest (K) and smallest (S) subfamilies. Also, overall, categorization of CsbZIPs should be linked to Arabidopsis bZIPs to some extent in the main texts.
In Fig.7, subgroups should be also indicated. It could be informative to understand the links between subgroups and functions.
Authors should analyze expression of selected bZIPs under heat stress. Authors mentioned the functional diversity of bZIPs in the introduction and discussion. Also, sensitivity of C. ensifolium to cold and heat stress is mentioned in the beginning of discussion part.
Reviewer 2 Report
Comments and Suggestions for Authors
Section 2.1, how reliable are the 70 predicted bZIPs? The authors use HMM to predict bZIPs in C. ensifolium, why choose this strategy? Is this the most accurate strategy to predict bZIPs? The author should test the reliability of this method in well-known plants such as Arabidopsis to see how many bZIPs are predicted correctly.
Section 2.2, please add the reason in the text for choosing Arabidopsis when constructing the phylogenetic tree.
Section 2.7, does the author randomly pick the 9 genes to do qPCR validation? If not, why choose these genes? The authors use heatmap from TBtools to show the expression levels of bZIPs, please provide the detailed parameters in method part. Besides, please also provide detailed description of 2-CT method for calculation of relative gene expression, and also the raw and processed qPCR data in supplementary tables.
Comments on the Quality of English LanguageExtensive editing of English language is required. It is hard to list all the language errors, so I will take the first paragraph as one example.
1-Line36: “...and has a highly conserved bZIP domain...” to “...and have a highly conserved bZIP domain…”
2-Line37: “the basic region is situated at the” to “the basic region is located at the”, “situated at” is often used when describing the geographical location of something.
3-Line39: “through the fixed structure N-X7-R/K motifs” to “through the N-X7-R/K motifs”, “fixed structure” and “motifs” are redundant.
4-Line40: “comparatively less conserved” to “relatively less conserved”, with “comparatively” you’d expect to know what it’s compared to. In other words, you should only use “comparatively” when you are making a direct comparison, whereas “relatively” can be used in a more general sense.
5-Line41: “amino acid residues” to “amino acid”, “amino acid residues” and “amino acid” are different.
6-Line44: “homodimer” to “homodimers”.
7-Line46 and Line48, “cis-acting” to “cis-acting”.
Round 2
Reviewer 1 Report
Comments and Suggestions for Authors
After the revision, the manuscript was much improved and authors properly answer to my comments.
Author Response
Thank you so much for your acceptance.
Reviewer 2 Report
Comments and Suggestions for Authors
The qPCR result is too ideal for the following reasons:
Firstly, the authors randomly pick up 9 genes and all 9 genes get exact the same trend as in heatmap (Figure 8), except for bZIP43. However, the exact same trend for bZIP43 lies in raw qPCR data (Table S5).
Secondly, the qPCR using biological replicates, not technique replicates, but the error bar is too small (Figure 9).
Thirdly, the fold change between different time points is too large from qPCR result compared to predicted gene expression levels, for example, CebZIP24 (12h) is 12 times higher than CebZIP24 (24h), actually, the predicted expression levels between 12h and 24h is less than 2 times (112.54 vs 76.11).
Based on these evidences, I do not trust this qPCR result.
Comments on the Quality of English LanguageThere are still many errors, I just list part of them.
Line25, “Nine-teen”, Nineteen?
Line26, “Cis-acting analysis”, what is Cis-acting analysis?
Line53, “AB13” should be ABI3
Line55, “In addition, AtbZIP18 interacts with AtbZIP34, and AtbZIP52 interacts with AtbZIP61”, these are proteins, should not use italics.
Line62, “Furthermore, bZIP TFs are involved in abiotic stresses, including low-temperature stress, drought stress, and salinity stress, as well as in biotic stresses, such as disease and pathogen defense”, this sentence does not have any reference?
Line65, “Arabidopsis thaliana”, but in line69, only “Arabidopsis”, they are not consistent.
Line72, “tomato”, but in line73, “tomatoes”, they are not consistent.
Line75, “5 ℃” should be “5℃”.
Line96, “The smallest CebZIP protein was CebZIP23 with 77 aa and 234 bp, while the largest was CebZIP67 (697 aa and 97 2094 bp)”, should be “The smallest CebZIP protein was CebZIP23 (77 aa and 234 bp), while the largest was CebZIP67 (697 aa and 97 2094 bp)”.
Line187, “cis-acting” should be “cis-acting”.
Line205, “transcriptome data”, Line207, “transcriptomic data”, not consistent.
Supplementary Table 5, “Calculation of relative gene expression formulae” should be “Calculation of relative gene expression formula”. And there is no data for reference gene.
Line391, “the GAPDH gene as the house-keeping gene (Table S4).”, I do not find GAPDH in Table S4.
Round 3
Reviewer 2 Report
Comments and Suggestions for Authors
Line393, “Three biological replicates and three technical replicates were performed.” Please also include the data of the technical replicates into Table S5.
Comments on the Quality of English LanguageI only checked abstract part and found 2 errors, there may be many other errors in the manuscript, please check it carefully.
Line19, “bioinformatics analysis” to “bioinformatic analysis”.
Line28, “universealy” to “universally”.
Round 4
Reviewer 2 Report
Comments and Suggestions for Authors
Thanks for your effort in improving this paper.
Comments on the Quality of English LanguageLine64,"AB13" should be "ABI3".